The effect of cigarette smoking habits on the outcome of dental implant treatment

Twito Dror Drortw99@gmail.com
Sade Paul
Department of Oral Rehabilitation, Medical Corps, IDF , Tel-Hashomer , Israel
Elangovan Satheesh
Electronic publication date: 2014 Sep 2
Publication date: 2014
Volume: 2
Electronic Location ID: e546
Received 2014 Jan 28; Accepted 2014 Aug 7
Copyright: © 2014 Twito and Sade
Copyright year: 2014
Copyright holder: Twito and Sade
License: This is an open access article distributed under the terms of the Creative Commons Attribution License, which permits unrestricted use, distribution, and reproduction in any medium, provided the original author and source are credited.
License URL: https://creativecommons.org/licenses/by/3.0/

Keywords: Implant failure, Implant survival, Tobacco

Funding: The authors declare there was no funding for this work.

==============================
The aim of this study was to analyze the influence of smoking habits and other possibly relevant factors on dental implant survival. The study population included all patients who underwent dental implants between the years 1999 and 2008 at a large military dental clinic and were examined in the periodic medical examination center.

Correlation between implant characteristics and patients’ smoking habits, as mentioned in the questionnaire answered by patients in the periodic examination, was performed.

Besides standard statistical methods, multiple linear regression models were constructed for estimation of the relative influence of some factors on implant survival rate. The long-term results of the implant treatment were good. The study refers to 7,680 implants. 7,359 (95.8%) survived and 321 (4.2%) did not survive. Concerning smoking habits, in a uni-variable analysis, factors found to have an association with implant survival were the smoking status of the patients (smoking/no smoking), the amount of smoking, passive smoking, and the time elapsed in ex-smokers from the time they ceased smoking to the time of implantation. In a multi-variable analysis, factors found to have an association with implant survival were smoking status (smoking/no smoking) and amounts of smoking as expressed in pack years.

Introduction

The success and predictability of implants are well established. For a great number of dental implant systems survival rates are within the 90 percentile (Buser et al., 1997; Adell et al., 1990; Naert et al., 2002; Vehemente et al., 2002; Jemt, Lekholm & Adell, 1989; Lindh et al., 1998; Brocard et al., 2000; Weibrich et al., 2001; Lekholm et al., 1999).

The success rates of implants tend to be lower than implant survival rates and change in relation to the measured criteria (implant mobility, bone loss, the presence of signs and symptoms, the resulting level of aesthetics, etc.). Despite the fact that implant survival and success rates are high, there is a growing impression that there are risk factors exposing patients to complications and ultimately to failure of implants. Among the perceived risks are occlusal overload, lower bone quality, and systemic diseases.

Smoking

A survey of health status in the United States has established a clear link between smoking and low levels of periodontal health and support (Ismail, Burt & Eklund, 1983; Schenkein et al., 1995). Tobacco smoking reduces leukocyte activity and is responsible for a low chemotactic migration rate, low mobility and low phagocytic activity (Krall & Dawson-Hughes, 1991). These impairments cause low infection resistance and interrupted wound healing (Krall & Dawson-Hughes, 1991). Smoking is also related to low calcium absorption (Krall & Dawson-Hughes, 1991). Reports in the literature show lower survivability of dental implants in smokers (Bain & Moy, 1993; Lambert, Morris & Ochi, 2000; De Bruyn & Collaert, 1994). One possible mechanism by which smoking might affect osseointegration is lowering blood flow rate due to increased peripheral resistance and platelet aggregation (Herzberg, Dolev & Schwartz-Arad, 2006; Levin & Schwartz-Arad, 2005). Smoking by-products such as CO and cyanide delay wound healing and together with nicotine inhibit cell proliferation (Levin & Schwartz-Arad, 2005). Tobacco directly affects osteoblast function (Levin & Schwartz-Arad, 2005). Strietzel et al. (2007) performed a systematic review and meta-analysis including 35 studies in order to analyze whether smoking affects implant prognosis with/without augmentations (Strietzel et al., 2007). In this review it was reported that 5 retrospective studies and one prospective study showed a relationship between smoking and inflammation around implants (peri-implantitis) (Strietzel et al., 2007). In 12 out of 13 studies there appeared to be significant marginal bone absorption in smokers compared to non-smokers (Strietzel et al., 2007).

Patients are usually advised to quit smoking at least two weeks before implant surgery in order to allow recovery of normal blood viscosity and platelet adhesion. Abstention from smoking should be extended at least 8 weeks after the implantation in order to permit the healing phase of the osteoblasts to take place (Bain & Moy, 1993).

The purpose of this research was to evaluate the relationship between smoking habits and the amount of smoking on dental implant survival.

The null hypothesis is that there is no relationship between the amount of smoking and implant survival.

Materials and Methods

The retrospective study was based on a consecutive cohort of patients who were treated by dental implants between the years 1999 and 2008 at a large military dental clinic, and were examined in a periodic medical examination center. No exclusion criteria applied. Two specialized oral and maxillofacial surgeons placed all implants. All patients underwent periodic checkups in the department of prosthodontics. Implant characteristics were recorded during these checkups. In these checkups the implant status was defined either as survival or failure. Survival was defined when an implant was satisfactorily functioning with no evidence of pain, suppuration, or inflammation. Otherwise it was defined as failure. Data on smoking among the study population is based on information gathered at the periodic medical examination center which conducts examinations for military personnel. A periodic medical examination is mandatory for the entire population of career soldiers from the age of thirty. Those aged thirty to thirty-four are required to perform the test once, from the age of thirty-five every three years, and every two years from the age of fifty. Smoking is reported using a self-administered questionnaire regarding health-related habits given as part of the periodic examination.

Patients who visited the periodic medical examination center no longer than a year before or after their implant placement were included in this study.

The following implant characteristics were recorded for those treated by dental implant (Table 1): implanted jaw, implanted region, immediate implantation, bone augmentation in conjunction to implant placement, sinus lift in conjunction to implant placement, membrane use, immediate loading, implant failure. Based on the self-administered questionnaire, the following smoking characteristics were recorded for those treated by dental implant (Table 2): smoking status, smoking years, number of cigarettes per day, pack years (calculated by multiplying the number of packs of cigarettes smoked per day by the number of years the person has smoked), past smokers- number of cigarettes per day, past smokers- number of years without smoking until implantation, exposure to passive smoking (the smoke of others) in closed places (home or at work) as qualitative data (exposure vs. no exposure).

Table 1 Implant properties.

Implant characteristics recorded for those treated by dental implant.

		Total N (%)	Failed
implants
N (%)	P	
Immediate loading	NO	7,628 (99.3)	316 (4.1)	0.065	
	YES	52 (0.7)	5 (9.6)		
Bone augmentation	NO	7,106 (92.5)	296 (4.2)	0.828	
	YES	574 (7.5)	25 (4.4)		
Membrane use	NO	7,481 (97.4)	314 (4.2)	0.857	
	YES	199 (2.6)	7 (3.5)		
Implanted jaw	MAXILA	3,160 (41.1)	95 (3)	0.007	
	MANDIBLE	3,761 (49)	174 (4.6)		
	MISSING INFORMATION	759 (9.9)			
Implanted region	MAX-front	834 (10.9)	33 (4.0)	0.0001	
	MAX-premolar	1,788 (23.3)	50 (2.8)		
	MAX-molar	538 (7.0)	12 (2.2)		
	MAN-front	400 (5.2)	46 (11.5)		
	MAN-premolar	969 (12.6)	43 (4.4)		
	MAN-molar	2,392 (31.1)	85 (3.6)		
	MISSING INFORMATION	759 (9.9)			

Table 2 Relationship between smoking habits and survivability of dental implants.

Implant failure rate was higher among smokers. There is a tendency to a higher failure rate with the increasing number of cigarettes per day.

Smoking habit		Failed
implants
N (%)	Surviving
implants
N (%)	p	
Smokers	YES	135 (5.6)	2,271 (94.4)	0.001	
	NO	185 (3.5)	5,074 (96.5)		
Number of cigarettes per day	1–10	32 (4.3)	720 (95.7)	0.059	
	11–20	50 (5.3)	891 (94.7)		
	21–30	34 (6.9)	462 (93.1)		
	31–40	18 (9.2)	177 (90.8)		
Smoking years	Non-smokers	185 (3.5)	5,074 (96.5)	0.001	
	Up to 10 years	14 (3.5)	384 (96.5)		
	More than 10 years	121 (6)	1,887 (94)		
Ex-smokers—no. of cigarettes per day	1–30	40 (2.8)	1,382 (97.2)	0.007	
	More than 30	20 (6.0)	311 (94.0)		
Ex-smokers—no. of years
without smoking until implantation	2–15	31 (2.3)	1,316 (97.7)	0.001	
Exposure to passive smoking
(the smoke of others) in closed places	Yes	39 (5)	741 (95.0)	0.001	
	No	21 (2.1)	994 (97.9)		
Smoking status	Non smokers	125 (3.6)	3,334 (96.4)	0.001	
	Past smokers	60 (3.3)	1,740 (96.7)		
	Present smokers	135 (5.6)	2,271 (94.4)		
Pack years	Non-smokers	186 (3.5)	5,088 (96.5)	0.001	
	x ≤ 1	15 (3.6)	399 (96.4)		
	1 < x ≤ 5	27 (4.3)	602 (95.7)		
	5 < x ≤ 10	41 (5.7)	676 (94.3)		
	x > 10	52 (8.0)	594 (92.0)		

Correlation between implant characteristics and patients’ smoking habits, as mentioned in the questionnaire, was performed.

The Ethics Committee of the Israel Defense Force Medical Corps approved the study (approval number: IDF - 879-2009).

Statistical analysis

Distribution of all the variables were tested. The relationship between each of the variables and implant failures was analyzed using a Chi-Square test and Fisher’s exact test. The difference in age between the surviving and failed implants was tested using a t-test. The correlation between smoking and implant failure was also tested in regression logistic models and OR and 95% CI were calculated. Four models were built: two univariable and two multi-variable models. In the uni-variable models, the dependent variable was implant failure and the independent variable used was once smoking status and once smoking groups by pack years. Multi-variable models were built in order to test the association between smoking and implant failure after neutralizing the effects of gender, age, type of loading, type and location of the implant. These variables were tested in the first block and entered in the model by the Stepwise Forward Selection (Likelihood Ratio) method with entry testing based on the significance of the score statistic, and removal testing based on the probability of a likelihood-ratio statistic based on the maximum partial likelihood estimates.

Entry = 0.05, removal = 0.10, maximum iterations 20. The following smoking variables: in the first model–smoking status and in the second model–smoking groups by pack years, entered the second block model.

Statistical significance was defined when p ≤ 0.05. Data was analyzed using the SAS and SPSS version 17 software.

Results

The study population consisted of 9,706 implants performed between the years 1999 and 2008. Information about smoking habits was available for 7,680 implants.

Implant properties: (Table 1)

The study refers to 7,680 implants. 7,359 (95.8%) survived and 321 (4.2%) did not survive. 6,731 implants (87.6%) were placed in men and 949 (12.4%) in women. Patients’ ages ranged from 22 to 55 years, averaging 41.48 years, with a standard deviation of 5.621, and a median of 42 years.

Relationship between smoking habits and survivability of dental implants (Table 2)

Implant failure rate was higher among smokers, 135 out of 2,406 (5.6%), compared to nonsmokers, 185 out of 5,259 (3.5%), p < 0.001. The figures show a tendency to a higher failure rate with the increasing number of cigarettes per day, 32 out of 752 (4.3%) among smokers of 1–10 cigarettes a day and up to 18 out of 195 (9.2%) among those who smoked 31–40 cigarettes a day, p = 0.059. There is a significant difference in relation to the number of cigarettes smoked per day in present smokers, between failed implants among those who smoked up to 30 cigarettes a day compared to those who smoked more than 30 cigarettes a day, p = 0.043 (Fig. 1).

Figure 1 Present smokers—failure percent by number of cigarettes.

A significant difference in relation to the number of cigarettes a day, in present smokers, between failed implants, among those who smoke up to 30 cigarettes a day compared to those who smoke more than 30 cigarettes a day.

A significant correlation was found between the implant failure rate and the following variables: smoking years, exposure to passive smoking in nonsmokers, smoking status and to the number of pack years (Figs. 2–5).

Figure 2 Present smokers—failure percent by years of smoking.

Significant correlation was found between implant failure rate and years of smoking.

Figure 3 Failure percent in relation to passive smoking.

Significant correlation was found between implant failure rate and exposure to passive smoking in non smokers.

Figure 4 Failure percent by smoking status.

Significant correlation was found between implant failure rate and smoking status.

Figure 5 Present smokers—failure percent by pack year.

Significant correlation was found between implant failure rate and the number of Pack year.

Correlation between implant survival and smoking in a uni- and multivariable analysis

In order to test the link between smoking habits alone and implant survival after neutralization of factors that were also found to have an impact on implant survival, two indices representing the phenomenon of smoking were selected: smoking status and pack years.

Smoking status

In the multivariable analysis presented in Table 3, it can be seen that even after neutralizing the effect of other variables (type of loading, type of implant rehabilitation and implanted jaw), present smoking clearly increases the risk of implant failure—P = 0.001, OR = 1.512.

Pack years

(1) The univariable analysis (Table 4) shows that smoking over five pack years increases significantly the chance of failure (OR = 1.659 in smokers of 5–10 pack years and OR = 2.395 in the group smoking over 10 pack years) compared to non-smokers.

(2) In a multi-variable analysis (Table 5), after neutralizing variables such as immediate implantation, type of prosthesis and implanted jaw, it was found that in two groups, pack years between 5 and 10 (OR = 1.683) and pack years greater than 10 (OR = 2.296), the risk of failure increases significantly in relation to non-smokers.

Table 3 Smoking status—multivariable analysis.

After neutralizing the effect of other variables (type of loading, the type of implant and rehabilitation), present smoking clearly increases the risk of implant failure P = 0.001, OR = 1.512.

	OR	95% CI	P	
Non Immediate implantation	1			
Immediate implantation	3.104	[1.179–8.171]	0.022	
Fixed prosthesis	1			
Removable prosthesis	2.212	[1.219–4.015]	0.009	
Maxilla	1			
Mandible	1.636	[1.264–2.119]	<0.001	
Smoking status				
Non smoker	1			
Past smoker	0.845	[0.617–1.159]	0.297	
Present smoker	1.512	[1.175–1.944]	0.001	
Notes.

Age and gender were found to be unrelated to implant failure and therefore were not entered in the model.

Table 4 Pack year uni-variable analysis.

Smoking over 5 pack years increases significantly the chance of failure (OR = 1.659 in smokers 5–10 pack year and OR = 2.395 in the group smoking over 10 pack years) compared to non-smokers.

Pack year	OR	95% CI	P	
Non smokers	1			
x ≤ 1	1.028	[0.602–1.757]	0.918	
1 < x ≤ 5	1.227	[0.812–1.853]	0.331	
5 < x ≤ 10	1.659	[1.172–2.349]	0.004	
x > 10	2.395	[1.741–3.294]	<0.0014	

Table 5 Pack year multi-variable analysis.

In 2 groups, Pack year between 5 and 10 (OR = 1.683) and pack year greater than 10 (OR = 2.296) the risk of failure increases significantly in relation to non smokers.

	OR	95% CI	P	
Non immediate implantation	1			
Immediate implantation	3.649	[1.383–9.630]	0.009	
Fixed prosthesis	1			
Removable prosthesis	2.035	[1.120–3.698]	0.20	
Maxilla	1			
Mandible	1.614	[1.274–2.089]	<0.001	
Pack Year				
Non smokers	1			
x ≤ 1	1.044	[0.610–1.788]	0.875	
1 < x ≤ 5	1.167	[0.768–1.772]	0.470	
5 < x ≤ 10	1.683	[1.187–2.387]	0.003	
x > 10	2.296	[1.662–3.172]	<0.0013	

Discussion

It is difficult to evaluate the role, importance and impact of a single risk factor in evaluating the results of treatment with implants. Chances are that there are a number of factors affecting implant such as bone quality, location in the mouth, type of prosthesis, para-functional habits, as well as systemic factors such as smoking, osteoporosis, genetic factors and more. The reality is that a large number of patients present a number of risk factors. The purpose of the study was to evaluate the relationship between smoking habits and the amount of smoking on dental implant survival. In relation to smoking habits, in a uni-variable analysis, factors found to have an association with implant survival are smoking status (smoking/no smoking), the amount of smoking, passive smoking, and the time elapsed in past smokers from the time of ceasing smoking to the time of implantation. In a multi-variable analysis, factors found to be in association with implant survival are smoking status (smoking/no smoking), and amount of smoking as expressed in pack years. Other factors found associated with implant survival, in relation to implant properties in a uni-variable analysis are immediate implantation, type of prosthesis (fixed/removable) and implanted jaw.

The significant difference found in relation to the type of prosthesis (fixed/removable) between those failed and surviving implants is in harmony with the literature. Based on an extensive literature review, Goodacre et al. (2003) indicate that in both arches, implant failure is higher in a removable prosthesis than on a fixed one.

Concerning the failure rate of the two arches, this research indicates a larger percentage of failures in the front of the mandible. Generally, but not always (Patrick et al., 1989), we find the failure rates to be higher in the maxilla (Goodacre et al., 2003; Jemt et al., 1996). It is possible that the reason for this higher rate of failure lies in the combination of the type of bone and time of failure. Most of the failures in this study occurred in the first year after implantation. There are reports in the literature that show a link between early failures and D1 bone type (Truhlar et al., 1994). The highest percentage of type D1-dense cortical bone is in the anterior mandible. This bone type is made histologically from a dense lamellar calcified bone which can stand high occlusal loading forces. However, this type of bone (D1) has fewer blood vessels in relation to other types of bone and therefore is dependent on the periosteum for the blood supply needed for healing the bone after placing the implant. The cortical bone receives all external blood supply in the outer external one-third of its width from the periosteum (Chanavaz, 1995; Vaughan, 1970) and therefore requires a delicate and minimal reflection during surgery. The various results linking smoking and the survival rate of implants correlates well with the reports in the literature (Bain & Moy, 1993; Lambert, Morris & Ochi, 2000; De Bruyn & Collaert, 1994). Similar to this study, which found more failures among smokers (5.6%) compared to non-smokers (3.5%), (Klokkevold & Han, 2007) report, after a statistical analysis of fourteen studies that dealt with the topic of dental implant survival rates, that the survival rate is lower by 2.68% in smokers than non-smokers. After performing a meta-analysis of 21 studies, Hinode et al. (2006) found that the risk of implant failure in smokers is higher significantly than in non-smokers—OR = 2.17 in smokers versus OR = 1.67 in non-smokers. However, data referring to the amount or quantity of smoking (No. of cigarettes or pack years) and its relation to implant survival rate is scarce. Indeed, Klokkevold & Han (2007) conclude at the end of a statistical analysis of 14 studies that an assessment of the relationship between the level and amount of smoking and implant survival rate cannot be done. Bain (2003) indicates that moderate smoking (11–20 cigarettes a day) and heavy smoking (over 20 cigarettes) raise the risk of implant failure, but only in a machined implant surface. In this study, a dose-related relationship between the amount of smoking and implant survival rate was found. This relationship is expressed in the following variables tested:

1. Number of cigarettes a day—in an analysis of uni-variables a significant difference was found regarding the number of cigarettes a day in present smokers. It turns out that smokers of over 30 cigarettes a day have a bigger chance of failure, up to 1.6 greater than those who smoke under 30 cigarettes a day.

2. Smoking years—present smokers who have smoked more than 10 years have a 1.7 times greater risk for implant failure than smokers who have smoked less than 10 years.

3. Pack years—the risk of failure rose significantly in 2 groups: the first, pack years of 5–10 (OR = 1.68) and second, pack years greater than 10 (OR = 2.296).

Another important issue in this study concerns passive smoking. A number of studies indicate a link between passive smoking and cardiovascular disease (Kritz, Schmid & Sinzinger, 1995; Barnoya & Glantz, 2005; Venn & Britton, 2007). Seemingly, an exposure to a mixture of toxins, chemicals, and carcinogens react with different mechanisms in the body and cause vascular damage, including endothelium inflammation, development of arthrosclerosis, protein and cytokine changes as well as induced platelet aggregation. Activities of the above mechanisms occurring alone or in combination with each other, act synergistically to develop cardiovascular diseases (Vardavas & Panagiotakos, 2009). So far there is no reference in the literature to the relation between passive smoking and implant failure. In this research a significant correlation was found between exposure to passive smoking and implant survival rate. The risk of implant failure among those exposed to passive smoking is 2.3 times bigger than the risk of those who are not exposed to passive smoking. This finding is in addition to the impact of passive smoking on cardiovascular disease mentioned above. It is possible that the factors at work are similar in both cases.

A correlation was found in this study between the number of years intervening between the cessation of smoking and time of implantation in ex-smokers, with a 2.7 times greater risk of implant failure in those who underwent the implantation up to 2 years from the time of cessation of smoking compared to those who had the implantation more than 2 years after the time of cessation. The reference in the literature to the protocol of cessation of smoking before implantation is first made by Bain (1996). Bain (1996) notes a significant difference between implant failure in those who followed the protocol in comparison to those who continued smoking. The protocol rule is to stop smoking one week before and eight weeks after placing the implants. This protocol is based on the medical literature showing improvement in blood circulation after one week of cessation of smoking and a histological proof of initial osseointegration taking place in the first eight weeks after implantation. As evidenced by the data arising from this study, it is possible that the survival rate rises as more time passes from cessation of smoking to implantation beyond the week advised by Bain’s protocol.

This study has several limitations that should be taken into consideration when evaluating the present results. First, using a retrospective cohort design makes it difficult to control for all other factors, which might differ between the different groups (smokers/non-smokers), passive smoking, and pack years. These and other factors are known as confounding variables. We dealt with this issue using the multi-variable models which were built in order to test the association between smoking and implant failure after neutralizing the effects of other variables (such as gender, age, type of loading, type and location of the implant, etc.) In addition, our study is biased by the fact that the subjects are a military population, which is a rather healthy population, above a certain age and come to the periodic medical examination center following army standing orders. Furthermore, relying on pre-existing self-reported data as we did in relying on the self-administered questionnaire limits the study in that the health-related habits are hard to verify independently. With regard to information on passive smoking, it is important to remark that although the data collected was of a qualitative nature (exposure vs. no exposure) and did not give a quantitative indication to the exposure to passive smoking in closed spaces, it allowed for the first time to relate to this important variable. Finally, using the implant as the unit of analysis, because of the nature of the data collection, makes it difficult to adjust for effects of clustering of outcomes within patients. Applying a different or more robust methodology might address the research problem more effectively in any future study. Despite these limitations, the study shows that after neutralizing the other variables, smoking status (smokers/non-smokers), passive smoking, and pack years are the variables relevant to implant survival.

Conclusion

The purpose of this research was to evaluate the relationship between smoking habits and dental implant survival.

The main findings in this research are that smoking status (smokers/non- smokers), passive smoking, and pack years are the variables found to be relevant to implant survival after neutralizing other variables. Thus, the null hypothesis was refuted.

The clinical outcomes of this study are related to a number of aspects:

1. As patients, smokers and passive smokers must be informed of the increased risk of failure.

2. Prevention—it is the obligation of the clinician treating a patient to guide and encourage the patient to cease smoking before undergoing implantation.

3. Treatment planning—the significant correlation found between smoking 5–10 pack years and greater than 10 pack years in implant failure should drive the clinician to search for alternative treatment plans, not including implants, that would have better chances of surviving in smoking patients.

Supplemental Information

Supplemental Information 1 Raw data

Click here for additional data file.

Additional Information and Declarations

Competing Interests

Author Contributions

Human Ethics

The authors declare there are no competing interests.

Dror Twito conceived and designed the experiments, performed the experiments, analyzed the data, contributed reagents/materials/analysis tools, wrote the paper, prepared figures and/or tables, reviewed drafts of the paper.

Paul Sade conceived and designed the experiments, performed the experiments, contributed reagents/materials/analysis tools, wrote the paper, reviewed drafts of the paper.

The following information was supplied relating to ethical approvals (i.e., approving body and any reference numbers):

Ethics Committee of IDF: approval number IDF - 879-2009.

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
