# Peer review of "The effect of cigarette smoking habits on the outcome of dental implant treatment"

_PeerJ, doi:10.7717/peerj.546_

## Round 0.1 · original submission · Major Revisions

This is a retrospective study confirming the effect of smoking on implant success.The study has several deficiencies that need to be addressed by the authors before it can be accepted for publication.

Reviewer 1 ·

Basic reporting

See attached doc

Experimental design

See attached doc

Validity of the findings

See attached doc

Additional comments

See attached doc

Annotated reviews are not available for download in order to protect the identity of reviewers who chose to remain anonymous.

Reviewer 2 ·

Basic reporting

The manuscript demonstrates findings of a retrospective study investigating the correlation of smoking to implant failure. It provides an additional evidence in the literature for the effect of amount of smoking and number of additional years of smoking.
Comments:
Introduction:
-Page 1 Line 12,13. "Similar findings were reported in a study that established the relation between smoking and periodontal support" may be omitted or summarized with the previous sentence.
- Page 1 Line 14,15. “Tobacco smoking reduces leukocyte activity and is responsible for a low chemotactic migration rate, low mobility and low phagocytic activity”. This needs to be referenced
- Page 1 Line 19-21. “It seems that smoking interferes with the process of
osseointegration by lowering blood flow rate due to increasing peripheral resistance
and platelets aggregation.” This is a possible mechanism that is hypothesized to be the reason for how smoking might affect osseointegration negatively. There is currently no firm evidence that this is the pathway of how smoking affect osseointegration. I would suggest rephrasing to indicate that one possible mechanism by which smoking might affect osseointegration is lowering blood flow rate due to increased peripheral resistance and platelet aggregation”.
-Page 1 Line 21-22. “Smoking by-products such as: CO and cyanide delay
wound healing and together with nicotine inhibit cell proliferation”. Please add a reference.
-Page 1 Line 22-23: “Tobacco affects directly the osteoblast function.” With reference 18. Please substitute this reference. The reference used should be an in vitro study demonstrating such an effect rather than a book reference.

-Page 1 Line 23. Strietzel et al study is not included in the list of references.

-Page 2 Line 2-4. “when in 12 out of 13 studies appeared to be significant marginal bone absorption in smokers compared to non-smokers”. This sentence needs to be rewritten. It is not clear. In addition, it seems it is not clear whether this statement was related to Strietzel et al study as the reference is relating to Heitz Mayfield. Please correct.

Methods:

Page 2 Line 18-19. “All military personnel are required to visit the periodic medical examination center every four years for medical examination that includes also a questionnaire regarding
health related habits.” Tense needs to be changed to past.

Page 3 Line 3. Please indicate what does IDF refer to?

The materials and methods section in its current format does not provide the reader with enough background to the variables that were investigated. The materials and methods section needs to be more detailed. The authors would need to describe the implant characteristics investigated and what type of correlations were made in relation to smoking (eg. Number of cigarettes, number of years, etc.).

Results:

Page 4. Line 5. “6731 implants (87.6%) were treated in men and 949 (12.4%) in women.” Please substitute treated with placed.

Page 4.Line 6-7. “age of implantation”. It is not clear to the reader what is meant by this phrase. Is that patient’s age or number of years since the implants were placed? Please clarify. The reviewer is not sure why the standard deviation will be relevant here.

-Table 1:
a) It is not clear whether bone augmentation/sinus lift were done prior or in conjunction to implant placement. Please clarify.
b) The title for this table should be more detailed to include a summary of the content of the table as well as what the p-value relates to. What type of statistical test was done here?

Page 4. Line 14. p= 0.059. it is not clear to the reader what type of statistical test was done here. What is considered the control here? Same applies to table 2.

Page 4. Line 14-17. “There is a significant difference in relation to the number of cigarettes a day, in present smokers, between failed implants, among those who smoke up to 30 cigarettes a day compared to those who smoke more than 30 cigarettes a day (Figure 1)”. No figure is present in the pdf supplied. What is the p-value here? This statement needs clarification. Some level of uncertainty is present here especially with the previous sentence stating that p-value was 0.059 (not significant) with higher number of cigarettes. I would suggest a more detailed description of what the previous p-value relates to and a clearer explanation of the significance present (what was tested/compared).

Page 4.Line 19. How can passive smoking be quantified. What about environmental exposure to smoke and how it might be relevant here.

Page 4. Line 18-20. A significant correlation was found between implant failure rate and the following variables: smoking years, exposure to passive smoking in non smokers, smoking
status and to the no' of packyear (Figures 2-5.) what was the p-value? No figures present. Please describe what is meant by no’ of packyear.

Page 5.Line 6-9. In The multivariable analysis presented in table 3 it can be seen that even after
neutralizing the effect of other variables (type of loading, the type of implant and rehabilitation). Please correct to the type of implant rehabilitation and implanted jaw.

Discussion:

First paragraph. Sentences are not well correlated. Needs to be rewritten with a better flow.
Page 7. Line 3. is in the front of the mandible. Replace front of the mandible with is in the anterior mandible.
Page 7. Line 4. “This bone type made histologicaly from a dense lamellar calcified bone can stand high occlusal loading forces”. Add is after type and which after bone to read as This bone type is made histologicaly from a dense lamellar calcified bone which can stand high occlusal loading forces.
Page 7. Line 6. “Therefore is depended”. Replace depended with dependent.”periost – replace with periosteum. Same correction in Line 8.
Page 7.Line 9-11. There is a repetition here.

Page 9. Line 15. Baine’s replace with Bain’s.

References: they do not follow the journal’s guidelines.

Experimental design

The materials and methods are briefly described. A more detailed description is needed to the variables studied. In addition, the data appears confusing in terms of what statistical tests were done at each point.

Validity of the findings

See comments earlier.

Additional comments

Several grammatical and spelling errors present.
The study addresses a very important risk factor that has been correlated to implant failures. This study would add to the list of evidence present in the literature and defines the potential effects of number of smoking years and amount of smoking which are not assessed deeply in the current evidence.

Reviewer 3 ·

Basic reporting

Dear Editor,
I have reviewed the manuscript titled “The Effect of Cigarette Smoking habits on the Outcome of
Dental Implant Treatment” which was submitted for publication in Peer J. The manuscript addresses an important topic. The following concerns need to be addressed by the authors before this manuscript is further considered for publication:
Introduction section: The manuscript sets up the context of the study well. I would recommend that the authors clearly state the hypothesis

Experimental design

Methods section: It will be helpful for readers if the authors describe in greater detail the various outcome and independent variables examined in the current study. In the statistical analysis section, I would recommend authors mentioning the type of regression model fit for examining the association between the independent and outcome variables. Were any model fit diagnostics done for the regression model? For example, for the logistic regression models, was Hosmer and Lemeshow Goodness of fit test statistic done?
Based on the analysis done, it appears that the outcome variable (survived or not) was used as a dichotomous variable. Did the authors consider using a survival analysis?

Validity of the findings

It is not clear to me what the unit of analysis is. Is it implant or the patient? If implant is the unit of analysis, how did the authors adjust for effects of clustering of outcomes within patients?
Tables 6 to 10 and none of the figures were available in the manuscript that was submitted for review. I would like to review these

Are the study results generalizable and externally valid? Please comment on this and the limitations of the study considering the retrospective nature of the data analysis.

---

## Round 0.2 · Minor Revisions

We appreciate the extensive modifications done by the authors.

Here are some of the minor issues that I noted:
1. The limitations section should go to the discussion rather than in conclusion.
2. Also, I still noticed several grammar errors. Please have it reviewed by the professional proof reader.
Examples: "blood supply needed for curing bone past implantation" and "delicate and minimal reflection is needed in time of surgery"

---

## Round 0.3 · accepted · Accept

Authors addressed all the comments.